# Associations of adverse maternal experiences and diabetes on postnatal maternal depression and child social-emotional outcomes in a South African community cohort

Yael K. Rayport[1,2], Ayesha Sania[1,2], Maristella Lucchini[1,2], Carlie Du Plessis[3], Mandy Potter[3], Priscilla E. Springer[4], Lissete A. Gimenez[1,2], Hein J. Odendaal[3], William P. Fifer[1,2,5], Lauren C. Shuffrey[1,2]*

1 Department of Psychiatry, Columbia University Irving Medical Center, New York, NY, United States of America, 2 Division of Developmental Neuroscience, New York State Psychiatric Institute, New York, NY, United States of America, 3 Department of Obstetrics and Gynaecology, Faculty of Medicine and Health Science, Stellenbosch University, Cape Town, Western Cape, South Africa, 4 Department of Paediatrics and Child Health, Stellenbosch University, Cape Town, Western Cape, South Africa, 5 Department of Pediatrics, Columbia University Irving Medical Center, New York, NY, United States of America

* lcg2129@cumc.columbia.edu

**Data Availability Statement:** Data from this study are available in the Dryad Digital Repository:

## Abstract

Previous literature has identified associations between diabetes during pregnancy and postnatal maternal depression. Both maternal conditions are associated with adverse consequences on childhood development. Despite an especially high prevalence of diabetes during pregnancy and maternal postnatal depression in low- and middle-income countries, related research predominates in high-income countries. In a South African cohort with or without diabetes, we investigated associations between adverse maternal experiences with postnatal maternal depression and child social-emotional outcomes. South African mother-child dyads were recruited from the Bishop Lavis community in Cape Town. Participants consisted of 82 mother-child dyads (53 women had GDM or type 2 diabetes). At 14–20 months postpartum, maternal self-report questionnaires were administered to assess household socioeconomic status, food insecurity, maternal depressive symptoms (Edinburgh Postnatal Depression Scale (EPDS)), maternal trauma (Life Events Checklist), and child social-emotional development (Brief Infant Toddler Social Emotional Assessment, Ages and Stages Questionnaires: Social-Emotional, Second Edition). Lower educational attainment, lower household income, food insecurity, living without a partner, and having experienced physical assault were each associated with postnatal maternal depressive symptoms and clinical maternal depression (EPDS ≥ 13). Maternal postnatal depression, lower maternal educational attainment, lower household income, household food insecurity, and living in a single-parent household were each associated with child social-emotional problems. Stratified analyses revealed maternal experiences (education, income, food insecurity, trauma) were associated with postnatal maternal depressive symptoms and child social-emotional problems only among dyads with *in utero* exposure to diabetes. Women

Rayport, Yael K. et al. (2022), Maternal experiences, diabetes, postnatal maternal depression, and child social-emotional outcomes in a South African community cohort, Dryad, Dataset, https://doi.org/10.5061/dryad.kkwh70s73.

**Funding:** This research was supported by the Bill and Melinda Gates Foundation (WPF; https://www.gatesfoundation.org/). LCS is supported by a National Institute of Child Health and Human Development K99/R00 Pathway to Independence Award (K99HD103910; https://www.nichd.nih.gov/). The content is solely the responsibility of the authors and does not necessarily represent the official views of the National Institutes of Health. The funders had no role in the study design, data collection and analysis, decision to publish, or preparation of the manuscript.

**Competing interests:** The authors have declared that no competing interests exist.

with pre-existing diabetes or gestational diabetes in LMIC settings should be screened for health related social needs to reduce the prevalence of depression and to promote child social-emotional development.

## 1. Introduction

Metabolic disorders, such as gestational diabetes mellitus (GDM), have been linked to postnatal maternal depression in low- and middle-income countries (LMICs) [1, 2]. Research on postnatal maternal depression predominates in high-income-countries (HICs) despite depression having an especially high prevalence in LMICs. Rates of maternal depression in South Africa are as high as 35% [3]. Prior research has demonstrated that *in utero* exposure to diabetes has long-term consequences on child emotional, behavioral, cognitive, and metabolic outcomes [4–6]. Children born to mothers with diabetes are at increased risk for specific neurodevelopmental sequelae including impaired memory and attention [7], as well as risk for autism spectrum disorder [8]. Postnatal maternal depression is independently associated with adverse childhood social-emotional developmental outcomes [9, 10], suggesting that comorbid GDM and postnatal depression could have a compounding influence on child development.

Structural social determinants of health including socioeconomic inequities and trauma have also been implicated in prenatal maternal depression. A study examining South African pregnant women from a lower socioeconomic status (SES) community found that they were more likely to be depressed if they were food insecure [11], suggesting an additive effect of multiple structural risk factors on mental health. This study also reported that women were more likely to be food insecure if they experienced suicidal ideation, had more than three children, less education, lower income, history of mental illness, experienced a threatening life event, or had been exposed to intimate partner violence [11], suggesting an interplay between household food insecurity and other maternal experiences. In a rural population from Southwestern Ethiopia, pregnant women who are food insecure were four times more likely to experience psychological distress, including depression, anxiety, or suicidality [12]. In developing countries, structural and socioeconomic inequities have linked to the risk of developing psychiatric disorders [13, 14]. Traumatic life events, such as intimate partner violence, are also associated with maternal depression. For example, in Khayelitsha, a township in the Western Cape Province of South Africa, the prevalence of depression ranged from 32–47% antenatally and 16–32% postnatally with intimate partner violence being a significant predictor of depression [15].

Structural social determinants of health including socioeconomic inequities and trauma have also been implicated in adverse childhood outcomes. Preschool aged children in food insecure households have lower literacy, numeracy, short-term memory, and self-regulation [16]. Previous research has also established associations between maternal trauma and childhood social-emotional development. For example, in a South African birth cohort study, maternal posttraumatic stress disorder was associated with poorer infant fine motor and adaptive motor development at six months of age [17].

To our knowledge, no studies have examined the relationships among structural and socioeconomic determinants, diabetes during pregnancy, postnatal maternal depression, and child social-emotional development in a low resource setting [18]. In a high-risk cohort with or without diabetes, we hypothesized that specific structural and socioeconomic determinants experienced by mothers—food insecurity and trauma—would increase a mother's risk of

developing postnatal depression and in turn, these variables would be associated with delays in social-emotional development in their children. We hypothesized maternal diabetes would moderate these associations. Specifically, we hypothesized a stronger association between trauma and food insecurity with postnatal depression in mothers who have diabetes compared to women who do not.

## 2. Methods

### 2.1. Ethics statement

All procedures complied with the ethical standards of the Institutional Review Boards and ethics review committees at Stellenbosch University (N16-08-101 and N06-10-210) and the New York State Psychiatric Institute (5338). All participants provided informed written consent at both time points (prenatal and postnatal) before inclusion in the study. The cohort was intended to continue recruiting, however recruitment stopped due to the coronavirus pandemic.

### 2.2. Population

Between April 2018 and March of 2020, participants were recruited from the Bishop Lavis community at their Health Center or the Diabetes Clinic in Tygerberg Academic Hospital in the Western Cape Province, South Africa. This study was built on existing collaborations with the university through the Safe Passage Study [19]. Results from the Safe Passage study revealed high rates of depression and anxiety within the population: 55% of mothers had a prenatal Edinburgh postnatal depression score (EPDS) $\geq$ 13 and 19.17% of mothers had a State-Trait Anxiety Inventory (STAI) $\geq$ 40 [19].

In the present study, women were recruited at their first antenatal care visit. Inclusion criteria for mothers included: able to communicate fluently in Afrikaans or English, able to provide informed consent, planned to deliver at Tygerberg Hospital, pregnant with one fetus, maternal age was $\geq$ 18 years, gestational age at study entry 6–24 weeks, and no major congenital abnormalities in the fetus on the second trimester ultrasound scan. Participants were excluded for the following: planned abortion, planned relocation from the area prior to delivery, advice against participation from a health care provider, or skin lesion on the anterior abdominal wall that could hamper with electrode placement (for a separate study procedure).

A total of 108 pregnant women were consented to participate in the study. Of those, there was 1 intrauterine death case, 4 infant demises, 7 study withdrawals, and 7 participants were lost of follow up. Of the remaining 89 participants, 7 of the women had type I diabetes and were excluded from the present analysis. The final sample consisted of 82 infant-mother dyads: N = 53 women with GDM or type 2 diabetes and N = 29 women without diabetes. Diabetic status was abstracted from maternal medical charts. Data from this study is available in the Dryad Digital Repository: https://doi.org/10.5061/dryad.kkwh70s73 [20].

### 2.3. Structural and socioeconomic risk factors

Maternal sociodemographic and socioeconomic information was acquired through study-specific maternal self-report surveys when the children were 14–21 months of age (16 ± 1.7 months). Sociodemographic factors included education (completion of primary school or less, some high school, to high school and beyond), and partner status (married or cohabitating with a partner versus not). Socioeconomic variables included household income reported in South African Rand (R) (0–900 R, 901–5000 R, 5001–10,000 R, to more than 10,000 R per month) and food insecurity (in the last 4 months, participants have 1) enough of the kind of

food they want, or 2) enough, but not always the kind of food they want, or 3) sometimes or often not enough to eat). Information on maternal history of trauma was acquired through the Life Events Checklist (LEC), a 15-item self-report measure which gathers information on whether a person has experienced potentially traumatic events [21]. The LEC has previously been used to assess PTSD in a perinatal population in South Africa [17]. Variables of trauma analyzed included whether the participant experienced physical assault (i.e., being attacked, hit, slapped, kicked, or beaten up) by selecting "happened to me" on the checklist.

## 2.4. Self-reported maternal depressive symptoms

Mental Health Questionnaires were administered along with the maternal self-report surveys. Depressive symptoms were measured with the Edinburgh Postnatal Depression Scale (EPDS), a 10-item screening tool assessing depressive symptoms in perinatal women with a higher score indicating more depressive symptoms. The EPDS has been validated as a screening instrument in South Africa in English and Afrikaans [22]. Previous reports have used a cut off score of $\geq 13$ to indicate probable depression in perinatal South African women [23].

## 2.5. Toddler social-emotional assessments

Developmental assessments were conducted from 14–21 months of age (16 ± 1.7 months). Adjusted follow-up age ranged from 14–20 months of age (16 ± 1.5 months). The Brief Infant Toddler Social Emotional Assessment (BITSEA), a 42-item parental report, was used to screen for child social-emotional behavioral problems and delays in social-emotional competence [24]. The BITSEA has been validated as a screener in socioeconomically and ethnically diverse cohorts, as well as in children with autism spectrum disorder [25]. The Ages & Stages Questionnaires: Social-Emotional, Second Edition (ASQ:SE:2), a parent-completed assessment, aimed to assess child social-emotional competence across a variety of situations [26]. The ASQ is widely used in LMICs and has been validated as a screening instrument in South Africa in English and Afrikaans [27].

## 2.6. Statistical analysis

All statistical analyses were performed using R version 4.0. Primary analyses consisted of a series of minimally and fully adjusted linear regression models to examine the main effects of maternal diabetic status and structural and socioeconomic risk factors on postnatal maternal depressive symptoms. Minimally adjusted models consisted of several univariate regression analyses to examine the association of maternal diabetic status during pregnancy, educational attainment, income, partner status, food insecurity, and physical assault with postnatal maternal depressive symptoms while controlling for maternal age. Fully adjusted models consisted of multivariate linear regression models that examined the association of maternal diabetic status during pregnancy, educational attainment, income, partner status, food insecurity, and physical assault with postnatal maternal depressive symptoms while controlling for maternal age in a single model. However, due to the association between household income and physical assault, these variables were estimated in separate models. For all models, we report standardized regression coefficients (β) and the 95% confidence intervals (CI) for each main effect. Secondary analyses consisted of logistic regression analyses to estimate the main effects of each risk factor with clinically relevant maternal depression, defined as having an EPDS score ≥13. We report unadjusted and adjusted odds ratios (ORs) and CI of the ORs. Linear regressions were used to estimate β and CI for correlates of the BITSEA Problem domain score, BITSEA Competence domain score, and ASQ:SE:2 score. Minimally and fully adjusted models were run as stated above but adjusted for child sex and adjusted age at assessment instead of

maternal age. Tertiary analysis involved running all the previously stated regression models stratified by diabetic status.

# 3. Results

## 3.1. Cohort characteristics

The sample included 82 mother-infant dyads. The average age of the women was $32 \pm 5.1$ years. Based on an EPDS cutoff of $\geq 13$, $N = 30$ women met criteria for clinical levels of depression (36.58%). $N = 53$ women had GDM or type 2 Diabetes (65%). Most of the women had some high school education (54%). The greatest percentage of women had a monthly household income of 901–5000 R (45%). The predominant household food insecurity status was having enough, but not always the kind of food they wanted to eat (40%). Most of the women had a spouse or partner (63%). Many of the women had experienced physical assault (45%). Just over half of the children were female (57%). The average gestational age at birth was $38 \pm 2.0$ weeks. The average child age at the follow-up assessment was $16 \pm 1.7$ months and the average adjusted age was $16 \pm 1.5$ months (Table 1). Table 2 provides an overview of the demographic characteristics by clinical depression diagnosis (Table 2).

## 3.2. Association of structural, socioeconomic, and maternal health risk factors with postnatal maternal depressive symptoms

In the minimally adjusted model, an educational attainment of primary school or less ($\beta$ = 5.275, 95% CI = 1.010 to 9.540), lower levels of income (0–900 R: $\beta$ = 9.051, 95% CI = 4.497 to 13.605; 901–5000 R: $\beta$ = 4.713, 95% CI = 1.180 to 8.246), living without a spouse or partner ($\beta$ = 4.385, 95% CI = 1.674 to 7.096), higher levels of household food insecurity (enough, but not always the kind of food they want to eat: $\beta$ = 3.133, 95% CI = 0.529 to 5.736); sometimes or often not enough to eat: $\beta$ = 6.123, 95% CI = 3.296 to 8.950), and having experienced physical assault ($\beta$ = 3.777, 95% CI = 1.253 to 6.300) were each significantly associated with increased maternal depressive symptoms on the EPDS. In the minimally adjusted model, maternal diabetes during pregnancy was not significantly associated with increased maternal depressive symptoms ($p$-values > 0.05) (Fig 1).

In the fully adjusted model, an educational attainment of primary school or less ($\beta$ = 4.015, 95% CI = 0.048 to 7.981), living without a spouse or partner ($\beta$ = 3.377, 95% CI = 0.670 to 6.084), higher levels of household food insecurity (enough, but not always the kind of food they want to eat: $\beta$ = 3.017, 95% CI = 0.544 to 5.489); sometimes or often not enough to eat: $\beta$ = 4.376, 95% CI = 1.464 to 7.288), and having experienced physical assault ($\beta$ = 2.549, 95% CI = 0.306 to 4.791) were all significantly associated with increased maternal depressive symptoms on the EPDS. In the fully adjusted model, maternal diabetes during pregnancy and income were not significantly associated with increased maternal depressive symptoms ($p$-values > 0.05) (Fig 1).

## 3.3. Association of structural, socioeconomic, and maternal health risk factors with postnatal maternal depressive symptoms with clinical maternal depression

In the minimally adjusted model, an educational attainment of primary school or less (OR = 12.773, 95% CI = 2.225 to 108.055), an income level of 0–900 R (OR = 13.949, 95% CI = 1.833 to 171.907), living without a spouse or partner (OR = 3.408, 95% CI = 1.174 to 10.427), the greatest degree of household food insecurity (sometimes or often not enough to eat: OR = 4.358, 95% CI = 1.320 to 15.869), and having experienced physical assault

**Table 1. Demographic characteristics of participants.**

| | Overall |
|---|---|
| | **(N = 82)** |
| | **Mean (SD)** |
| | **or N (%)** |
| **Maternal Age (years)** | |
| Mean (SD) | 32 ± 5.1 |
| **Clinical Depression** | |
| **(EPDS ≥ 13)** | |
| Not clinically depressed | 52 (63) |
| Clinically depressed | 30 (37) |
| **Diabetic Status** | |
| Non-diabetic | 29 (35) |
| GDM or type 2 diabetes | 53 (65) |
| **Education** | |
| Primary school or less | 8 (10) |
| Some high school | 44 (54) |
| Completed high school or beyond | 30 (37) |
| **Monthly household income** | |
| 0–900 R | 9 (11) |
| 901–5000 R | 37 (45) |
| 5001–10,000 R | 26 (32) |
| More than 10,000 R | 10 (12) |
| **Food Security** (within the last 4 months, how would you describe the food in your household…) | |
| Enough of the kind of food we want | 26 (32) |
| Enough, but not always the kind of food we want to eat | 33 (40) |
| Sometimes or often not enough to eat | 23 (28) |
| **Partner** | |
| Does not live with a spouse or partner | 19 (23) |
| Lives with a spouse or partner | 63 (77) |
| **Experienced Physical Assault (i.e., being attacked, hit, slapped, kicked, beaten up)** | |
| Has not experienced | 38 (46) |
| Has experienced | 37 (45) |
| Missing | 7 (8.5) |
| **Child Sex** | |
| Male | 35 (43) |
| Female | 47 (57) |
| **Gestational Age at Delivery (weeks)** | |
| Mean (SD) | 38 ± 2.0 |
| **Child age at follow up Assessment (months)** | |
| Mean (SD) | 16 ± 1.7 |
| **Child adjusted Age at follow up Assessment (months)** | |
| Mean (SD) | 16 ± 1.5 |

(OR = 3.684, 95% CI = 1.317 to 11.143) were each significantly associated with clinical levels of maternal depression. In the minimally adjusted model, maternal diabetes during pregnancy was not significantly associated with clinical levels of maternal depression ($p$-$values > 0.05$) (Fig 1).

**Table 2. Demographic characteristics of participants across depression groups.**

| | Not Clinically Depressed (N = 52) Mean ± SD or N (%) | Clinically Depressed (N = 30) Mean (SD) or N (%) | Overall (N = 82) Mean (SD) or N (%) |
|---|---|---|---|
| **Maternal Age (years)** | | | |
| Mean (SD) | 32 ± 5.3 | 32 ± 4.8 | 32 ± 5.1 |
| **Maternal Diabetic Status** | | | |
| Non-diabetic | 20 (38) | 9 (30) | 29 (35) |
| GDM or type 2 diabetes | 32 (62) | 21 (70) | 53 (65) |
| **Maternal Education** | | | |
| Primary school or less | 2 (4) | 6 (20) | 8 (10) |
| Some high school | 26 (50) | 18 (60) | 44 (54) |
| Completed high school or beyond | 24 (46) | 6 (20) | 30 (37) |
| **Monthly household income** | | | |
| 0–900 R | 2 (4) | 7 (23) | 9 (11) |
| 901–5000 R | 25 (48) | 12 (40) | 37 (45) |
| 5001–10,000 R | 17 (33) | 9 (30) | 26 (32) |
| More than 10,000 R | 8 (15) | 2 (7) | 10 (12) |
| **Food Security** (within the last 4 months, how would you describe the food in your household…) | | | |
| Enough of the kind of food we want | 20 (38) | 6 (20) | 26 (32) |
| Enough, but not always the kind of food we want to eat | 22 (42) | 11 (37) | 33 (40) |
| Sometimes or often not enough to eat | 10 (19) | 13 (43) | 23 (28) |
| **Maternal Partner Status** | | | |
| Does not live with a spouse or partner | 8 (15) | 11 (37) | 19 (23) |
| Lives with a spouse or partner | 44 (85) | 19 (63) | 63 (77) |
| **Maternal Experience of Physical Assault** (i.e., being attacked, hit, slapped, kicked, beaten up) | | | |
| Has not experienced | 30 (58) | 8 (27) | 38 (46) |
| Has experienced | 19 (37) | 18 (60) | 37 (45) |
| Missing | 3 (5.8) | 4 (13.3) | 7 (8.5) |
| **Child Sex** | | | |
| Male | 21 (40) | 14 (47) | 35 (43) |
| Female | 31 (60) | 16 (53) | 47 (57) |
| **Gestational Age at Delivery (weeks)** | | | |
| Mean (SD) | 38 ± 2.1 | 37 ± 1.8 | 38 ± 2.0 |
| **Child Age at follow up Assessment (months)** | | | |
| Mean (SD) | 16 ± 1.7 | 17 ± 1.6 | 16 ± 1.7 |
| **Child adjusted Age at follow up Assessment (months)** | | | |
| Mean (SD) | 16 ± 1.5 | 17 ± 1.5 | 16 ± 1.5 |

In the fully adjusted model, lower levels of education (primary school or less: OR = 22.844, 95% CI = 2.128 to 387.973; some high school: OR = 6.340, 95% CI = 1.205 to 45.259), living without a spouse or partner (OR = 12.399, 95% CI = 2.218 to 103.842), and the greatest degree of household food insecurity (sometimes or often not enough to eat: OR = 6.936, 95% CI = 1.107 to 63.898) were all significantly associated with clinical levels of maternal depression. In the fully adjusted model, maternal diabetes during pregnancy, income, and experiencing physical assault were not significantly associated with clinical levels of maternal depression ($p$-values > 0.05) (Fig 1).

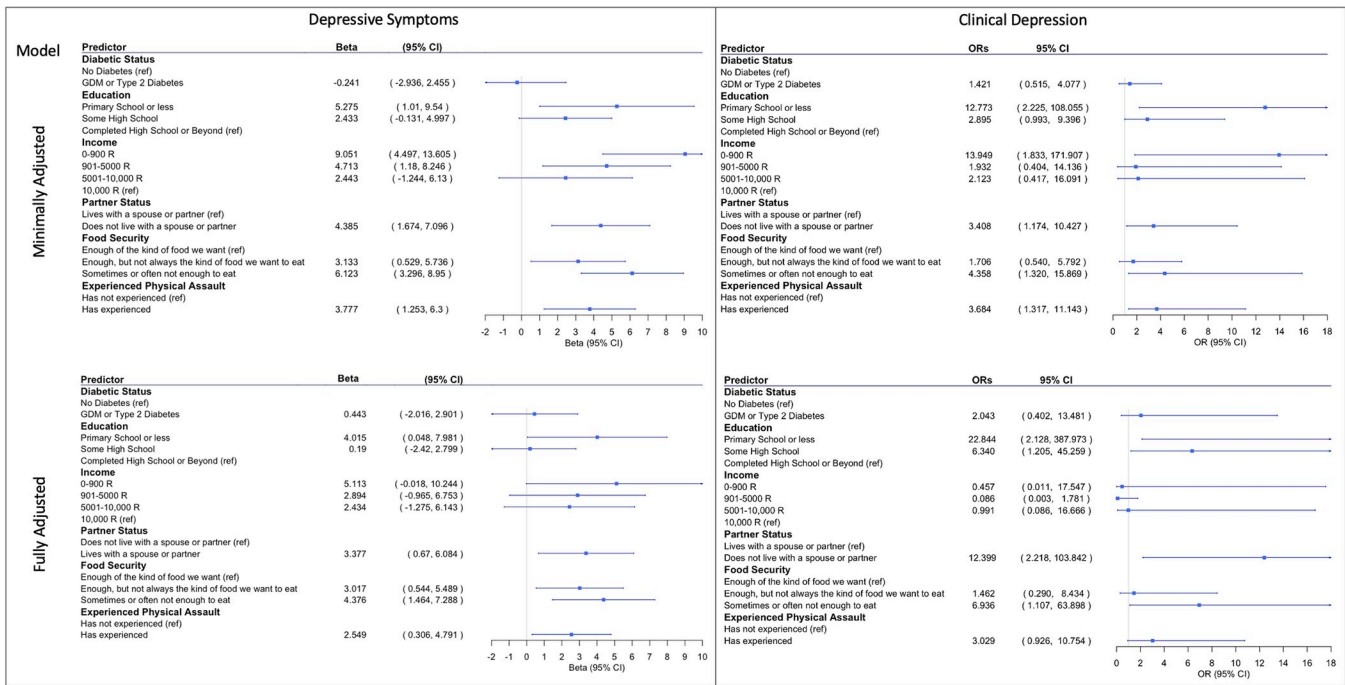

**Fig 1.** Betas and 95% CI for correlates of depressive symptoms (left) and odds ratios and 95% CI for correlates of clinical depression (EPDS > = 13) (right). Minimally adjusted model was adjusted for maternal age. Fully adjusted model was adjusted for maternal age as well as the other maternal experiences. Maternal physical assault and income were analyzed in separate models.

### 3.4. Association of structural, socioeconomic, and maternal physical and mental health risk factors with child social-emotional problems on the BITSEA

In the minimally adjusted model, clinical maternal depression (β = 3.043, 95% CI = 0.279 to 5.808) and a maternal educational attainment of primary school or less (β = 5.497, 95% CI = 0.737 to 10.256) were each associated with a higher BITSEA Problem Scores. Maternal diabetes during pregnancy, living without a spouse or partner, household food insecurity, and maternal physical assault were not associated with higher BITSEA Problem Scores in the minimally adjusted model (*p-values* > 0.05). In the fully adjusted model, household food insecurity (having enough, but not always the kind of food they want to eat) was also associated with increased BITSEA Problem Scores (β = 6.162, 95% CI = 0.849 to 11.475). Clinical depression, maternal diabetes during pregnancy, educational attainment, income, living without a spouse or partner, and maternal physical assault were not associated with higher BITSEA Problem Scores in the fully adjusted model (*p-values* > 0.05) (Fig 2).

### 3.5. Association of structural, socioeconomic, and maternal physical and mental health risk factors with child social-emotional competence on the BITSEA

In the minimally adjusted model, an education level of primary school or less (β = -2.208, 95% CI = -4.353 to -0.064) and lower levels of income (901–5000 R) (β = –2.312, 95% CI = -4.418 to -0.205) were each associated with decreased BITSEA Competence Scores. In the minimally adjusted model, maternal clinical depression, diabetes during pregnancy, household food insecurity, and maternal physical assault were not associated with decreased BITSEA Competence Scores (*p-values* > 0.05) (Fig 2).

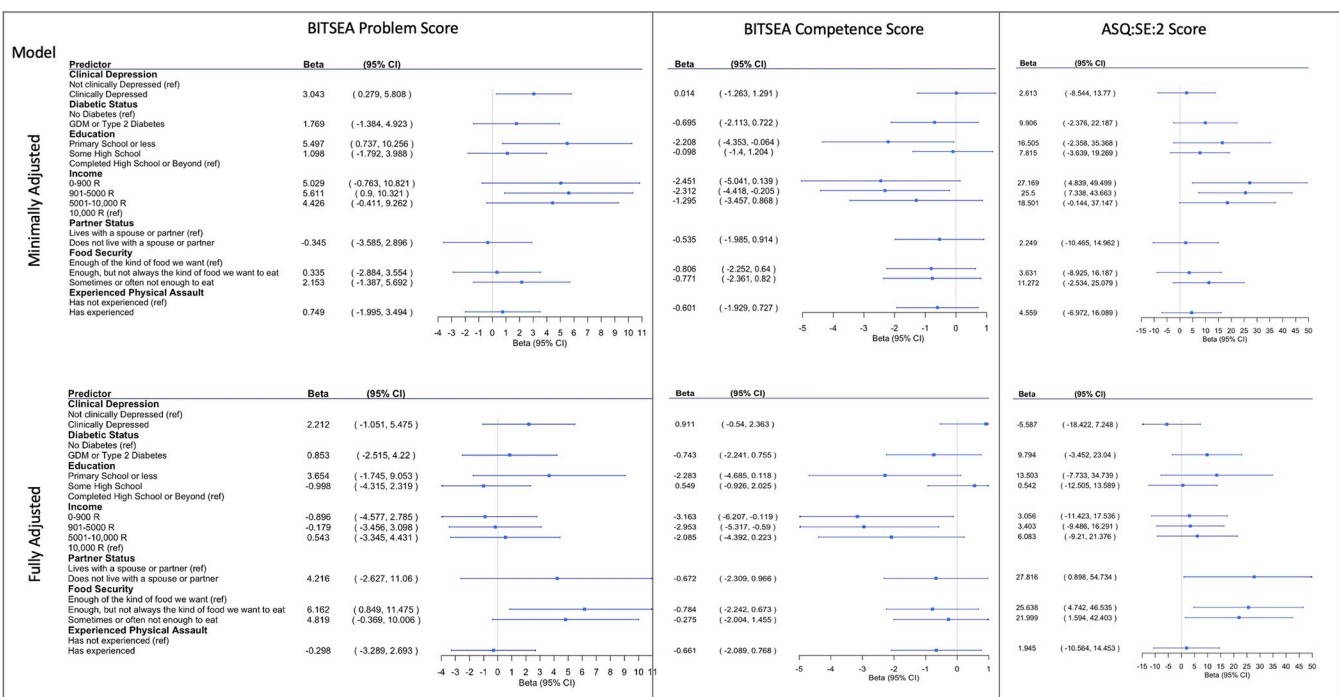

**Fig 2.** Betas and 95% CI for correlates of BITSEA Problem Score (left), BITSEA Competence Score (middle) and ASQ:SE:2 Scored (right). Minimally adjusted model was adjusted for maternal age. Fully adjusted model was adjusted for maternal age as well as the other maternal experiences. Maternal physical assault and income were analyzed in separate models.

In the fully adjusted model, income levels of 0–900 R ($\beta$ = -3.163, 95% CI = -6.207 to -0.119) and 901–5000 R per month ($\beta$ = -2.953, 95% CI = -5.317 to -0.590) were associated with were associated with decreased BITSEA Competence Scores. Maternal clinical depression, diabetes during pregnancy, living without a spouse or partner, household food insecurity, and maternal physical assault were not associated with decreased BITSEA Competence Scores ($p > 0.05$). In the fully adjusted model, maternal clinical depression, diabetes during pregnancy, educational attainment, household food insecurity, and maternal physical assault were not associated with decreased BITSEA Competence Scores ($p > 0.05$) (Fig 2).

### 3.6. Association of structural, socioeconomic, and maternal physical and mental health risk factors with child social-emotional development on the ASQ:SE:2

In the minimally adjusted model, lower levels of income (0–900 R) ($\beta$ = 27.169, 95% CI = 4.839 to 49.499) and 901–5000 R ($\beta$ = 25.500, 95% CI = 7.338 to 43.663) were associated with a higher ASQ:SE:2 scores, which are indicative of poorer social-emotional development. In the minimally adjusted model, maternal clinical depression, diabetes during pregnancy, educational attainment, living without a spouse or partner, household food insecurity, and maternal physical assault were not associated with higher ASQ:SE:2 scores ($p > 0.05$). In the fully adjusted model, living without a spouse or partner ($\beta$ = 27.816, 95% CI = 0.898 to 54.734) and household food insecurity were associated with higher ASQ:SE:2 scores (enough, but not always the kind of food we want to eat: $\beta$ = 25.638, 95% CI = 4.742 to 46.535; sometimes or often not enough to eat: $\beta$ = 21.999, 95% CI = 1.594 to 42.403). In the fully adjusted model, maternal clinical depression, diabetes during pregnancy, educational attainment, income, and maternal physical assault were not associated with higher ASQ:SE:2 scores ($p > 0.05$) (Fig 2).

### 3.7. Association of structural, socioeconomic, and maternal health risk factors with postnatal maternal depressive symptoms stratified by maternal diabetes status

In the minimally adjusted model, among women with diabetes during pregnancy, an educational attainment of primary school or less (β = 5.916, 95% CI = 1.488 to 10.343), lower levels of income (0–900 R: (β = 12.777, 95% CI = 7.142 to 18.413; and 901–5000 R β = 5.503, 95% CI = 1.910 to 9.097), household food insecurity (sometimes or often not having enough food to eat: β = 6.482, 95% CI = 3.161 to 9.804), and having experienced physical assault (β = 4.294, 95% CI = 1.103 to 7.485) were each associated with increased postnatal maternal depressive symptoms on the EPDS. Living without a spouse or partner was not associated with increased maternal depressive symptoms among women with diabetes in the minimally adjusted model (p > 0.05) (Fig 3).

In the fully adjusted model, among women with diabetes during pregnancy, an educational attainment of primary school or less (β = 4.599, 95% CI = 0.420 to 8.778), the lowest level of income (0–900 R: β = 6.827, 95% CI = 0.366 to 13.287), household food insecurity (Having enough food but not always the kind of food they want to eat: β = 3.355, 95% CI = 0.297 to 6.413; and Sometimes or often not having enough to eat: β = 5.060, 95% CI = 1.691 to 8.430), and having experienced physical assault (β = 3.712, 95% CI = 1.082 to 6.342) were associated with increased postnatal maternal depressive symptoms on the EPDS. In the fully adjusted model among diabetic women, living without a spouse or partner was not significantly associated with increased maternal depressive symptoms (p > 0.05) (Fig 3).

In the minimally adjusted model, among women without diabetes during pregnancy, living without a spouse or partner (β = 6.281, 95% CI = 2.440 to 10.121) was associated with

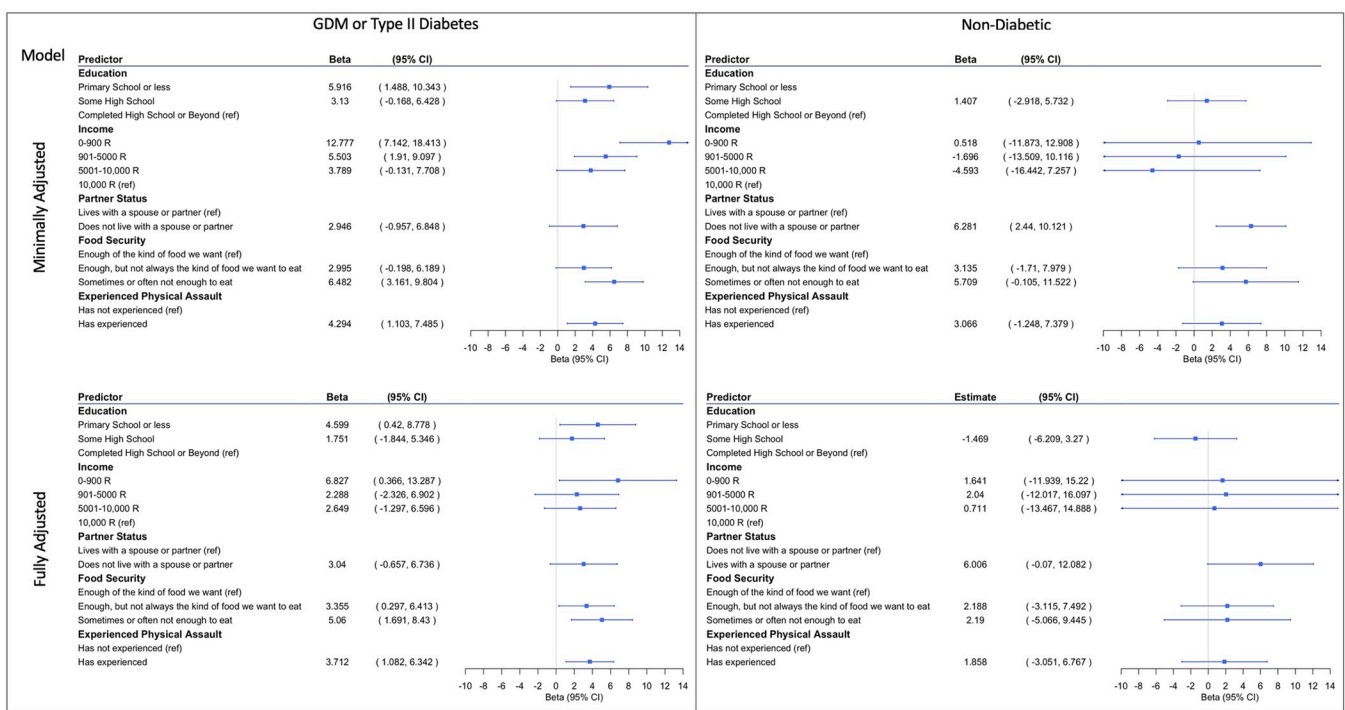

**Fig 3. Stratified by diabetes betas and 95% CI for correlates of depression.** Minimally adjusted model was adjusted for maternal age. Fully adjusted model was adjusted for maternal age as well as the other maternal experiences. Maternal physical assault and income were analyzed in separate models.

increased postnatal maternal depressive symptoms. Among, non-diabetic women, maternal educational attainment, income, household food insecurity, and experiencing physical assault were not significantly associated with increased postnatal maternal depressive symptoms in the minimally adjusted model. In the fully adjusted model, among women without diabetes during pregnancy, none of the maternal experiences were significantly associated with postnatal maternal depressive symptoms ($p > 0.05$) (Fig 3).

### 3.8. Association of structural, socioeconomic, and maternal physical and mental health risk factors with child social-emotional problems on the BITSEA stratified by maternal diabetic status

In the minimally adjusted model, among children with *in utero* exposure to diabetes, maternal educational attainment of primary school or less (β = 5.525, 95% CI = 0.035 to 11.016) and lower levels of household income (901–5000 R: β = 7.361, 95% CI = 1.933 to 12.788) were each associated with significantly higher BITSEA Problem Scores. Maternal clinical depression, living in a single-parent household, household food insecurity, and maternal trauma were not associated with significantly higher BITSEA Problem Scores (*p-values* > 0.05). In the fully adjusted model, household food insecurity (having enough but not always the kind of food they want to eat: β = 9.961 95% CI = 2.816 to 17.106; and sometimes or often not having enough to eat: β = 6.498, 95% CI = 0.071 to 12.925) was associated with higher BITSEA Problem Scores. Maternal clinical depression, maternal education, household income, living in a single-parent household, and maternal trauma were not associated with significantly higher BITSEA Problem Scores (*p-values* > 0.05) (Fig 4).

In the minimally adjusted model, among children without *in utero* exposure to diabetes, there were no significant associations between maternal experiences and BITSEA problem

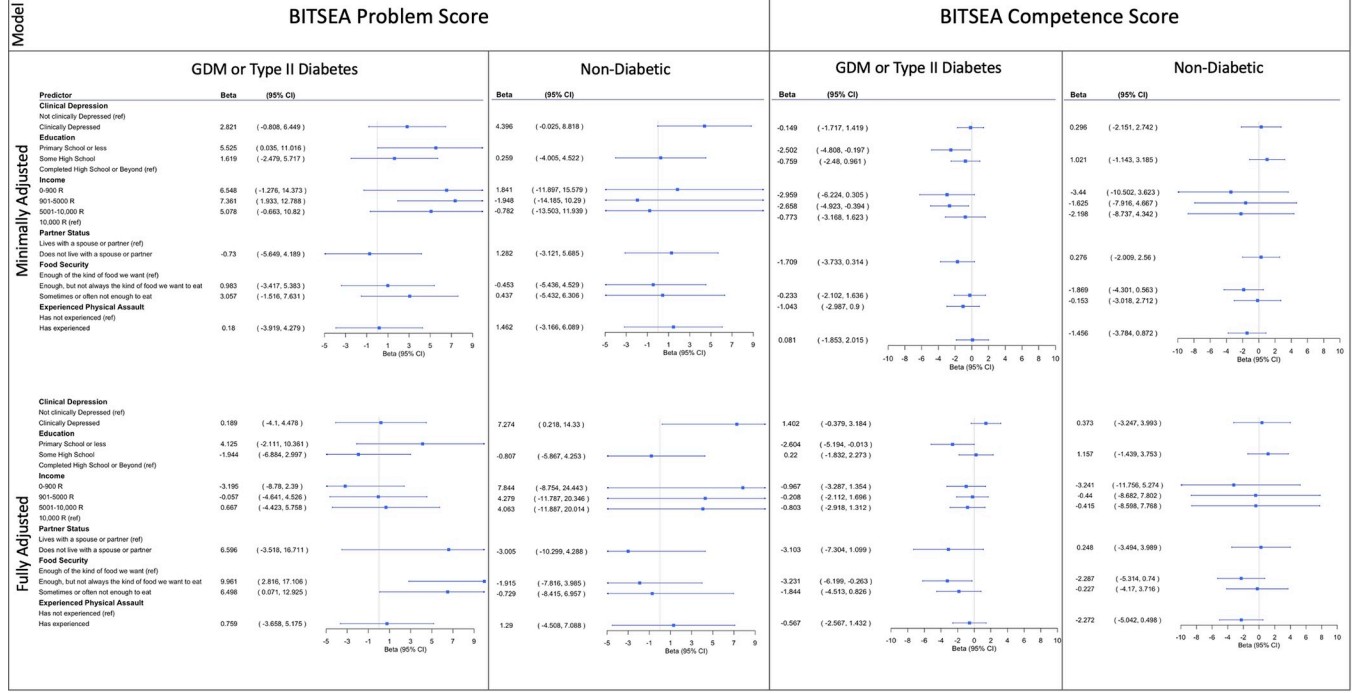

**Fig 4.** Stratified by diabetes betas and 95% CI for correlates of BITSEA Problem (left) and Competence Score (right). Minimally adjusted model was adjusted for maternal age. Fully adjusted model was adjusted for maternal age as well as the other maternal experiences. Maternal physical assault and income were analyzed in separate models.

scores. However, in the fully adjusted model, among children without *in utero* exposure to diabetes, clinical depression was a significant predictor of higher BITSEA Problem Scores ($\beta$ = 7.274, 95% CI = 0.218 to 14.330). Maternal education, household income, living in a single-parent household, household food insecurity, and maternal trauma were not associated with higher BITSEA Problem Scores (*p-values* > 0.05) (Fig 4).

### 3.9. Association of structural, socioeconomic, and maternal physical and mental health risk factors with child social-emotional competence on the BITSEA stratified by maternal diabetic status

In the minimally and fully adjusted model, among children with *in utero* exposure to diabetes, maternal educational attainment of primary school or less was associated with lower BITSEA Competence Scores (minimally adjusted: $\beta$ = -2.502, 95% CI = -4.808 to -0.197; fully adjusted: $\beta$ = -2.604, 95% CI = -5.194 to -0.013). In the minimally adjusted model only, a household income of 901–5000 R was associated with lower BITSEA Competence Scores ($\beta$ = -2.658, 95% CI = -4.923 to -0.394). In the fully adjusted model only, household food insecurity (having enough but not always the kind of food they want to eat: $\beta$ = -3.231, 95% CI = -6.199 to -0.263) was associated with lower BITSEA Competence Scores. In the minimally and fully adjusted models, maternal clinical depression, a single-parent household, and maternal trauma were not significantly associated with BITSEA Competence Scores (*p-values* > 0.05). Household food insecurity in the minimally adjusted model and household income in the fully adjusted model were not significantly associated with BITSEA Competence Scores (*p* > 0.05) (Fig 4).

Among children without *in utero* exposure to diabetes, none of the maternal experiences were associated with lower BITSEA Competence Scores in either the minimally or fully adjusted models (*p-values* > 0.05) (Fig 4).

### 3.10. Association of structural, socioeconomic, and maternal physical and mental health risk factors with child social-emotional development on the ASQ:SE:2 stratified by diabetes

In the minimally adjusted model, among children with *in utero* exposure to diabetes, lower household income (901–5000 R) was associated with significantly higher ASQ:SE:2 scores ($\beta$ = 27.974, 95% CI = 7.218 to 48.730). In the minimally adjusted model, maternal clinical depression, maternal educational attainment, living in a single-parent household, household food insecurity, and maternal trauma were not associated with ASQ:SE:2 scores (*p-values* > 0.05). In the fully adjusted model, household food insecurity (having enough, but not always the kind of food they want to eat) was associated with higher ASQ:SE:2 scores ($\beta$ = 35.677, 95% CI = 7.789 to 63.564). Maternal clinical depression, maternal educational attainment, household income, living in a single-parent household, and maternal trauma were not associated with ASQ:SE:2 scores (*p* > 0.05) (Fig 5).

Among children without *in utero* exposure to diabetes, none of the maternal experiences were associated with ASQ:SE:2 scores in either the minimally or fully adjusted models (*p* > 0.05) (Fig 5).

## 4. Discussion

### 4.1. Summary

This is the first study to evaluate specific structural and socioeconomic determinants experienced by mothers–educational attainment, household income, partner status, household food insecurity, and history of trauma–in association with postnatal maternal depression in a LMIC

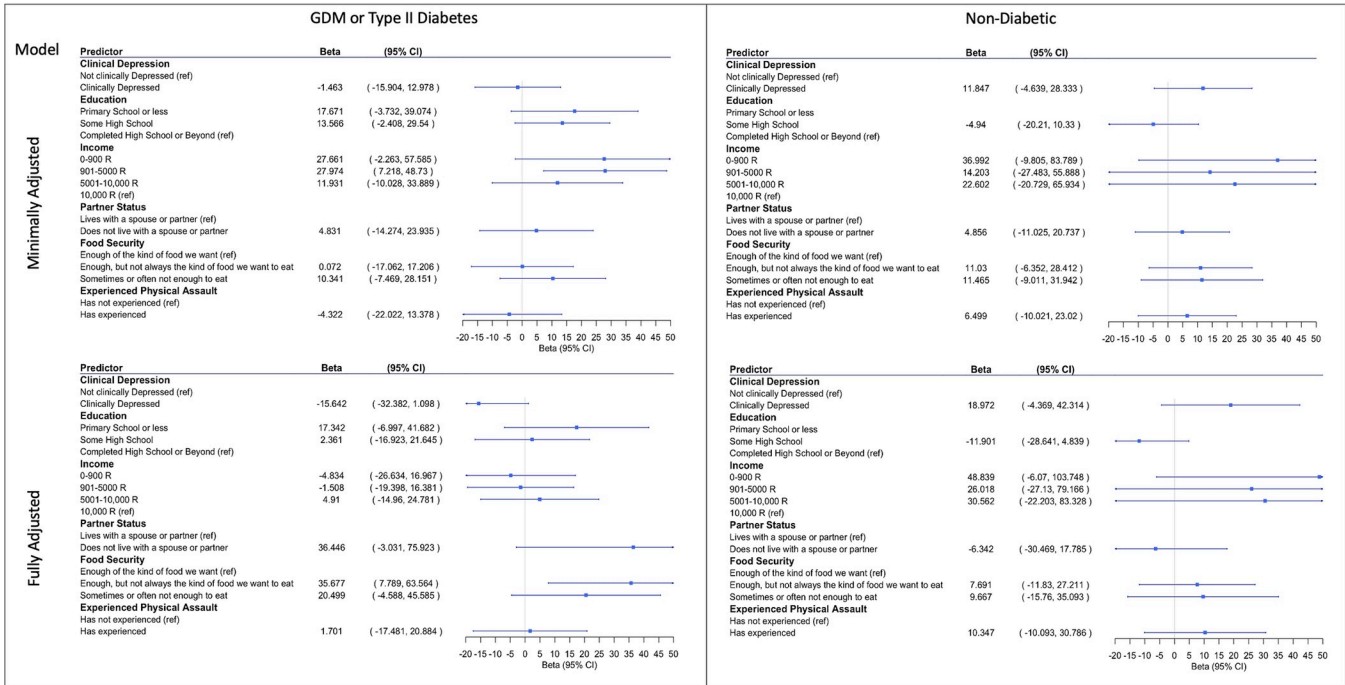

**Fig 5. Stratified by diabetes betas and 95% CI for ASQ:SE:2 score.** Minimally adjusted model was adjusted for maternal age. Fully adjusted model was adjusted for maternal age as well as the other maternal experiences. Maternal physical assault and income were analyzed in separate models.

cohort of women with or without diabetes during pregnancy. It is also the first to our knowledge to study social-emotional developmental outcomes of children born to these women. This cohort had a high prevalence of clinical depression at 36% (EPDS score ≥ 13). Many of the adverse maternal experiences were associated with increased postnatal maternal depressive symptoms and clinical levels of maternal depression based on dichotomized EPDS scores. Stratified analyses revealed maternal diabetic status significantly moderated these associations. Significant associations between specific structural and socioeconomic determinants experienced by mothers and postnatal maternal depression were only observed in women with diabetes during pregnancy. When examining associations among specific structural and socioeconomic determinants experienced by mothers and maternal diabetic status with childhood social-emotional outcomes, we demonstrated that many of the structural and socioeconomic determinants were significantly associated with worse child social-emotional outcomes among children with *in utero* diabetes exposure, and less so among those without *in utero* diabetes exposure.

## 4.2. Association of structural, socioeconomic, and maternal health risk factors with postnatal maternal depression

Predictors of maternal depressive symptoms and clinical depression included lower levels of education, lower levels of income, living without a partner or spouse, higher levels of household food insecurity, and having experienced physical assault. Prior studies exploring perinatal depression in South African women have also found that specific structural and socioeconomic determinants including education, income, marital status, food insecurity, stressful life events in the last 12 months, and intimate partner violence are predictive of perinatal depression [28–30]. Other studies have shown the interconnectedness of structural and

socioeconomic determinants experienced by mothers. For example, correlates of intimate partner violence experienced by mothers with depressive symptoms were associated with emotional distress and food insecurity [31]. Thus, structural and socioeconomic experienced by mothers may have an interactive effect on adverse maternal mental health outcomes.

In this study, we did not observe an association between diabetes during pregnancy and postnatal maternal depression. However, prior studies have demonstrated diabetes during pregnancy is a risk factor for postpartum depression [1, 2]. Possible explanations for our lack of replication include the small sample size of the cohort or heterogeneity of diabetes with both GDM and type 2 diabetes in the diabetes group. A systematic review examining diabetes during pregnancy and maternal depression pre-pregnancy, during pregnancy, or postpartum found different rates of depression depending on the diabetes type. The prevalence of depression among women with GDM was reported to range from 4.1% to 80% with a median of 14.7% whereas the prevalence of depression among women with pre-existing diabetes type 1 and type 2 diabetes) ranged from 0% to 60% with a median of 8.3% [32]. Future studies should examine the effects of diabetes on depression in a larger cohort with representation of each diabetes type in a LMIC setting.

### 4.3. Association of structural, socioeconomic, and maternal health risk factors with postnatal maternal depression stratified by maternal diabetes status

In this study, among women with diabetes, lower levels of education and income, household food insecurity, and experiencing physical assault were associated with increased postnatal maternal depressive symptoms and clinical depression. For non-diabetic women this relationship was not observed, as only living without a spouse or partner was associated with maternal postnatal depression. Given that most of the maternal experiences were associated with depression in the diabetic group only, diabetes could serve as a potential moderator between maternal experiences and depression. This finding suggests diabetes has a compounding effect on depression when comorbid with other adverse maternal experiences. In a larger cohort, we would conduct an interaction analysis to further support diabetes' role as a moderator.

### 4.4. Association of structural, socioeconomic, and maternal physical and mental health risk factors with child social-emotional development

This study also found that predictors of increased child social-emotional problems included clinical maternal depression, lower levels of maternal education, lower household income, and household food insecurity. Correlates of decreased behavioral competencies included lower levels of maternal education and lower household income. Correlates of worse social-emotional development included lower household income, household food insecurity, and living in a single-parent household.

Other studies have also demonstrated structural and socioeconomic factors as strong predictors of child outcomes. SES has been found to be most strongly associated with child cognitive development at 5 years of age [33], supporting this study's findings that education, income, and household food insecurity play the most influential role in child outcomes. However, we only observed an association between postnatal maternal depression and child social-emotional problems but not with social-emotional competencies. Similarly, the effects of depression on child development are mixed: one study has found that children of postnatally depressed mothers tend to be more aggressive, are hospitalized more, but have similar cognitive and social development to children born to non-depressed mothers [34]. A study

examining the development of children at 2 years of age found that infant outcomes were similar despite postnatal maternal depression [28].

Among children with *in utero* exposure to diabetes, lower levels of maternal education, lower household income, and household food insecurity were associated with higher social-emotional problems and lower social-emotional competencies; lower household income and household food insecurity were associated with worse social-emotional development. Among children without *in utero* exposure to diabetes, maternal depression was correlated with higher social-emotional problems.

### 4.5. Strengths and limitations

Strengths of this study include the range of maternal structural and socioeconomic determinants analyzed, the heterogeneity in maternal postnatal depression status, and the two assessments of child social-emotional development (the BITSEA and ASQ:SE:2). The primary limitations of this study include a lack of information regarding prenatal maternal mood disorders during pregnancy and the small sample size. The small sample size limited our ability to run a more sophisticated moderation analysis for diabetic status. Additionally, we were unable to group by diabetes type (GDM vs type 2) restricting our ability to differentiate the effects of each type of diabetes on postnatal depression. In the stratified analysis, the binary variable of clinical depression was not used because of the limited power in the small sample. Other limitations include the use of maternal report measures to examine child social-emotional development. Another limitation is that participants were recruited only from two clinics in the urban South African community. Therefore, the results may not be as generalizable to the population beyond pregnant women living in urban South Africa. Nevertheless, these results are an important contribution to the literature since few studies have investigated these associations in LMIC settings.

### 4.6. Conclusion and implications

Overall, the results of this study support our hypothesis that structural and socioeconomic determinants experienced by mothers, such as lower household income, household food insecurity, and experiencing trauma increase a woman's risk of developing postnatal depression and this relationship is influenced by maternal diabetes during pregnancy. SES factors, such as income, education, and food insecurity, as well as maternal depression also influence child emotional and behavioral development. This study suggests that women with pre-existing diabetes or gestational diabetes in LMIC settings should be screened for health related social needs to reduce the prevalence of depression and to promote child social-emotional development. Future studies should consider health related social needs as a target for intervention among pre-conception and perinatal women in LMIC settings.

## Acknowledgments

The authors would like to thank the mothers and children who participated in the study. We would also like to acknowledge Lucy Brink, J. David Nugent, Margaret Shair, Carmen Condon, and Daianna Rodriguez for their data management and administrative support. In addition, we wish to thank Dr. Diedré Mason at the Diabetic Clinic for her advice and support.

## Author Contributions

**Conceptualization:** Yael K. Rayport, Ayesha Sania, Maristella Lucchini, Priscilla E. Springer, Lissete A. Gimenez, William P. Fifer, Lauren C. Shuffrey.

**Data curation:** Yael K. Rayport, Carlie Du Plessis, Mandy Potter, Priscilla E. Springer, Lissete A. Gimenez, Hein J. Odendaal, Lauren C. Shuffrey.

**Formal analysis:** Yael K. Rayport, Ayesha Sania, Maristella Lucchini, Lauren C. Shuffrey.

**Funding acquisition:** Hein J. Odendaal, William P. Fifer, Lauren C. Shuffrey.

**Investigation:** Yael K. Rayport.

**Methodology:** Ayesha Sania.

**Project administration:** Carlie Du Plessis, Mandy Potter.

**Supervision:** William P. Fifer, Lauren C. Shuffrey.

**Visualization:** Yael K. Rayport.

**Writing – original draft:** Yael K. Rayport.

**Writing – review & editing:** Yael K. Rayport, Ayesha Sania, Maristella Lucchini, Carlie Du Plessis, Mandy Potter, Priscilla E. Springer, Lissete A. Gimenez, Hein J. Odendaal, William P. Fifer, Lauren C. Shuffrey.

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
