## [Decision Letter · Decision Letter 0]

11 Jun 2022

PGPH-D-22-00573

Adverse maternal experiences associated with maternal diabetes, postnatal maternal depression, and child social-emotional development in South African mother-child dyads

Dear Dr. Rayport,

Thank you for submitting your manuscript to PLOS Global Public Health. After careful consideration, we feel that it has merit but does not fully meet PLOS Global Public Health’s publication criteria as it currently stands. Therefore, we invite you to submit a revised version of the manuscript that addresses the points raised during the review process.

Please submit your revised manuscript by . If you will need more time than this to complete your revisions, please reply to this message or contact the journal office at globalpubhealth@plos.org. Please include the following items when submitting your revised manuscript:

We look forward to receiving your revised manuscript.

Kind regards,

Priyanka Baloni

Academic Editor

Journal Requirements:

1. In the online submission form, you indicated that your data will be submitted to the Dryad database upon acceptance. Should your submission be accepted, we will require the following information in your Data Availability Statement: 

a. The DOI provided by Dryad

b. The citation for your data package in the reference section of your manuscript

c. The citation for your data package in the methods section

If you are unable to adhere to our open data policy, please kindly revise your statement to explain your reasoning and we will seek the editor's input on an exemption. Please be assured that, once you have provided your new statement, the assessment of your exemption will not hold up the peer review process."

2. We do not publish any copyright or trademark symbols that usually accompany proprietary names, eg (R), (C), or TM  (e.g. next to drug or reagent names). Please remove all instances of trademark/copyright symbols throughout the text, including Questionnaires® on page 7.

Additional Editor Comments (if provided):

Reviewers' comments:

Reviewer's Responses to Questions

**Comments to the Author**

1. Does this manuscript meet PLOS Global Public Health’s publication criteria? Is the manuscript technically sound, and do the data support the conclusions? The manuscript must describe methodologically and ethically rigorous research with conclusions that are appropriately drawn based on the data presented.

Reviewer #1: Partly

Reviewer #2: Partly

2. Has the statistical analysis been performed appropriately and rigorously?

Reviewer #1: Yes

Reviewer #2: Yes

3. Have the authors made all data underlying the findings in their manuscript fully available (please refer to the Data Availability Statement at the start of the manuscript PDF file)?

Reviewer #1: No

Reviewer #2: Yes

4. Is the manuscript presented in an intelligible fashion and written in standard English?

Reviewer #1: Yes

Reviewer #2: Yes

5. Review Comments to the Author

Reviewer #1: Important elements are missing in the methods section. The authors also need to reorganize some paragraphs (specified in a separate file attached herewith) to improve coherence of ideas. They should also avoid redundancies.

Reviewer #2: I want to appreciate the authors for their effort and time in investing more in asses this cohort study in the context of South Africa. The Background, the result, and conclusions of this manuscript are well organized and addressed well but the method section and the discussion need to address and improve. In addition to this editorial errors in the paper, tables and figures also need to address. Here are some comments that need to be addressed.

Background (abstract section): It is better if you say something in the background instead of listing the objective only. Introduction: In the first paragraph you said that "To the best of our knowledge, there have been no studies ………" Remove this part (you stated on the last paragraph of the introduction).

Methods: This section is good but you failed to address deeply about the study area, sample size, selection criteria, and measuring tool (you addressed but not listed from where did you take (citation)) for the risk factors (Title Number 2.3). In the first section of the Methods Include briefly about the study area for readers. Put the Ethical issue as a subtitle and discuss about the ethical issue in the last section of the methods (including COVID-19). The way that you measured the economic status, physical assault, and food security status is not clear and not cited the reference. Why you did not use wealth index (PCAs))?

Result: The head of the tables needs editing and the percentage (%) written in the table is repeatedly written in the tables (you can write as N (%) at the top of the characteristics (for example Table 1). The way how that you measure the physical assault (as shown in the table) is not clear (how did you measure it)? The way that you measured the food security as shown in the table is also not clear. Discussion: This section is good but you did not discuss the pertinent findings of this study with the other findings (mainly the variables (the summary part)). Include abbreviations, ethics, data availability, funding, statement of conflict of interest, and author's contributions in the last section (above the reference). The figures (Fig 1-5) which are listed below the reference are not visible to read (I am unable to read this part).

6. PLOS authors have the option to publish the peer review history of their article (what does this mean?). If published, this will include your full peer review and any attached files.

**Do you want your identity to be public for this peer review?** For information about this choice, including consent withdrawal, please see our Privacy Policy.

Reviewer #1: **Yes: **Negalign Berhanu Bayou (PhD in Global Health, Associate Professor; Department of Health Policy and Economics, Institute of Health; Jimma University, Ethiopia; Postdoctoral Fellow, HaSET MCH Program, Harvard University, EPHI and St. Paul's Hospital MMC)

Reviewer #2: No

---

## [Decision Letter · Decision Letter 1]

9 Sep 2022

Associations of adverse maternal experiences and diabetes on postnatal maternal depression and child social-emotional outcomes in a South African community cohort

PGPH-D-22-00573R1

Dear Ms. Rayport,

We are pleased to inform you that your manuscript 'Associations of adverse maternal experiences and diabetes on postnatal maternal depression and child social-emotional outcomes in a South African community cohort' has been provisionally accepted for publication in PLOS Global Public Health.

Best regards,

Priyanka Baloni

Academic Editor

We thank the authors for addressing the reviewers comments and editing the manuscript. Due to unavailability of reviewers during the summer break, there was a delay in getting the reviews in and making a decision for your manuscript. Congratulations to you and the team for highlighting the factors associated with postnatal maternal depression.

Reviewer Comments (if any, and for reference):

Reviewer's Responses to Questions

**Comments to the Author**

1. If the authors have adequately addressed your comments raised in a previous round of review and you feel that this manuscript is now acceptable for publication, you may indicate that here to bypass the “Comments to the Author” section, enter your conflict of interest statement in the “Confidential to Editor” section, and submit your "Accept" recommendation.

Reviewer #2: All comments have been addressed

2. Does this manuscript meet PLOS Global Public Health’s publication criteria? Is the manuscript technically sound, and do the data support the conclusions? The manuscript must describe methodologically and ethically rigorous research with conclusions that are appropriately drawn based on the data presented.

Reviewer #2: Yes

3. Has the statistical analysis been performed appropriately and rigorously?

Reviewer #2: Yes

4. Have the authors made all data underlying the findings in their manuscript fully available (please refer to the Data Availability Statement at the start of the manuscript PDF file)?

Reviewer #2: Yes

5. Is the manuscript presented in an intelligible fashion and written in standard English?

Reviewer #2: Yes

6. Review Comments to the Author

Reviewer #2: The authors addressed all comments but still, there are some editorial comments that need to be addressed.

7. PLOS authors have the option to publish the peer review history of their article (what does this mean?). If published, this will include your full peer review and any attached files.

**Do you want your identity to be public for this peer review?** For information about this choice, including consent withdrawal, please see our Privacy Policy.

Reviewer #2: No
